# Brain-derived and in vitro-seeded alpha-synuclein fibrils exhibit distinct biophysical profiles

**Selene Seoyun Lee, Livia Civitelli\*[†], Laura Parkkinen\*[†]**

Nuffield Department of Clinical Neurosciences, Oxford Parkinson's Disease Center, University of Oxford, Oxford, United Kingdom

## eLife Assessment

This **important** work compares the strain properties of a-synuclein fibrils isolated from LBD and MSA patient samples with the resulting amplified fibrils following SAA. Using orthogonal biochemical and structural approaches to strengthen their analyses, the authors provide **solid** evidence that the SAA-amplified fibrils do not recapitulate the disease-relevant strains present in the patient samples. CryoEM would further strengthen this data but it is outside the scope of the work. This work should be considered in the widespread applications of SAA in synucleopathies and its potential limitations.

**\*For correspondence:**
livia.civitelli@ndcn.ox.ac.uk (LC);
laura.parkkinen@ndcn.ox.ac.uk (LP)

[†]These authors contributed equally to this work

**Competing interest:** The authors declare that no competing interests exist.

**Abstract** The alpha-synuclein (αSyn) seeding amplification assay (SAA) that allows the generation of disease-specific in vitro seeded fibrils (SAA fibrils) is used as a research tool to study the connection between the structure of αSyn fibrils, cellular seeding/spreading, and the clinicopathological manifestations of different synucleinopathies. However, structural differences between human brain-derived and SAA αSyn fibrils have been recently highlighted. Here, we characterize the biophysical properties of the human brain-derived αSyn fibrils from the brains of patients with Parkinson's disease with and without dementia (PD, PDD), dementia with Lewy bodies (DLB), multiple system atrophy (MSA), and compare them to the 'model' SAA fibrils. We report that the brain-derived αSyn fibrils show distinct biochemical profiles, which were not replicated in the corresponding SAA fibrils. Furthermore, the brain-derived αSyn fibrils from all synucleinopathies displayed a mixture of 'straight' and 'twisted' microscopic structures. However, the PD, PDD, and DLB SAA fibrils had a 'straight' structure, whereas MSA SAA fibrils showed a 'twisted' structure. Finally, the brain-derived αSyn fibrils from all four synucleinopathies were phosphorylated (S129). Interestingly, phosphorylated αSyn were carried over to the PDD and DLB SAA fibrils. Our findings demonstrate the limitation of the SAA fibrils modeling the brain-derived αSyn fibrils and pay attention to the necessity of deepening the understanding of the SAA fibrillation methodology.

## Introduction

Synucleinopathies are all characterized by abnormal accumulation of the protein αSyn in the brain. However, they are clinically and neuropathologically highly heterogeneous diseases with prominent disease-specific differences in the presentation of symptoms, rate of disease progression, and the brain regions and cell types are vulnerable to αSyn deposition and neuronal death. In PD, PDD, and DLB, αSyn aggregation is found in the neuronal soma as Lewy bodies (LBs) and in the axons and dendrites as Lewy neurites (LNs) (*Figure 1*, *Forno, 1988*; *Spillantini et al., 1997*; *Braak et al., 2003*). Furthermore, the astroglial accumulation of αSyn is a prominent but so far underdiagnosed pathology

**eLife digest** Alpha-synuclein is a protein that is essential for brain function. Like all proteins, alpha-synuclein is made of a string of amino acids and folded up in a precise way. Misfolding of this protein can lead to a buildup of protein clumps in the brain. These clumps cause neurodegenerative diseases like Parkinson's disease, Parkinson's disease with dementia, dementia with Lewy bodies, and multiple system atrophy.

Each of these diseases affects different parts of the brain and has distinct symptoms. Studying the misfolding patterns and structures of proteins collected from the brains of deceased people who had these conditions may help scientists understand the differences. Scientists also try to recreate these misfolded proteins in the laboratory using a seeding amplification assay technique. This is important as scientists can only extract limited amounts of protein clumps from human brains. However, it is unclear if the laboratory-produced protein clumps behave exactly like brain-derived clumps. Learning more is essential to make sure that studies of these proteins produce accurate results.

Lee et al. identified differences between alpha-synuclein clumps extracted from patients with different brain diseases. They also show that laboratory-derived protein clumps do not exactly recreate those from patients. Clumps from patients with Parkinson's disease, Parkinson's disease with dementia, dementia with Lewy bodies, and multiple system atrophy have straight and twisted protein structures when examined under an electron microscope. But the laboratory-generated clumps for Parkinson's disease, Parkinson's disease with dementia, and dementia with Lewy bodies are all straight. Laboratory-generated clumps for multiple system atrophy are all twisted. The clumps from patient brains also have many phosphate groups attached to them. Laboratory-derived clumps of Parkinson's disease with dementia and dementia with Lewy bodies also had this feature. However, the lab-derived versions of the other disease clumps lack this characteristic.

Lee et al. identify essential differences between the protein clumps from the brains of patients with different neurodegenerative diseases. They also found that the laboratory-generated clumps do not completely replicate all their features. More research is needed on these disease-specific differences and how they contribute to specific disease symptoms and disease progress. Work is also required to refine laboratory processes for protein clumps to replicate disease-associated clumps better. More studies could help scientists develop better laboratory models for these diseases that can be used to create and test new therapies.

in these Lewy body disorders (*Altay et al., 2022*). The αSyn pathology in MSA is primarily found in oligodendrocytes as glial cytoplasmic inclusions (GCIs) (*Papp et al., 1989*).

The mechanistic link between αSyn and clinicopathological diversity of the synucleinopathies is hypothesized to be the different fibrillar 'strains' of αSyn in analogy to prion disease. A strain is generated when αSyn monomers fold into specific fibrillar forms with distinct conformational and biological characteristics (*Melki, 2018*). Under the disease condition, misfolded and aggregated αSyn recruits normal, soluble endogenous αSyn to aggregate, and this self-perpetuating process spreads throughout the brain-periphery axis. Several studies have generated αSyn fibrils in vitro using various experimental conditions to characterize the presence of different strains (*Bousset et al., 2013*; *Peelaerts et al., 2015*; *Li et al., 2018*; *Guerrero-Ferreira et al., 2019*; *Lau et al., 2020*), but their relevance to the human condition remains questionable. However, some studies have also examined αSyn fibrils extracted from the human brain tissue of different synucleinopathies and these have demonstrated that the human-derived αSyn strains exhibit distinct structures (*Schweighauser et al., 2020*), infectivity and bioactivity in cells (*Woerman et al., 2018a*; *Woerman et al., 2019*; *Ayers et al., 2022*) and animals (*Prusiner et al., 2015*; *Lau et al., 2020*; *Holec et al., 2022*; *Peng et al., 2018*). The intrinsic structure of αSyn proteoforms may also depend on the local conditions of the brain region where they are formed and thus may vary within a single disease entity (*Peng et al., 2018*; *Lau et al., 2020*; *Schweighauser et al., 2020*). Future studies must comprehensively map the distinct fingerprints of human brain-derived αSyn strains to the disease.

SAAs have become more applicable due to their feasibility in amplifying in vitro-seeded αSyn fibrils (SAA fibrils) from human biosamples. SAA refers to two distinct seed amplification techniques: real-time quaking-induced conversion (RT-QuIC) and protein misfolding cyclic amplification (PMCA). Both

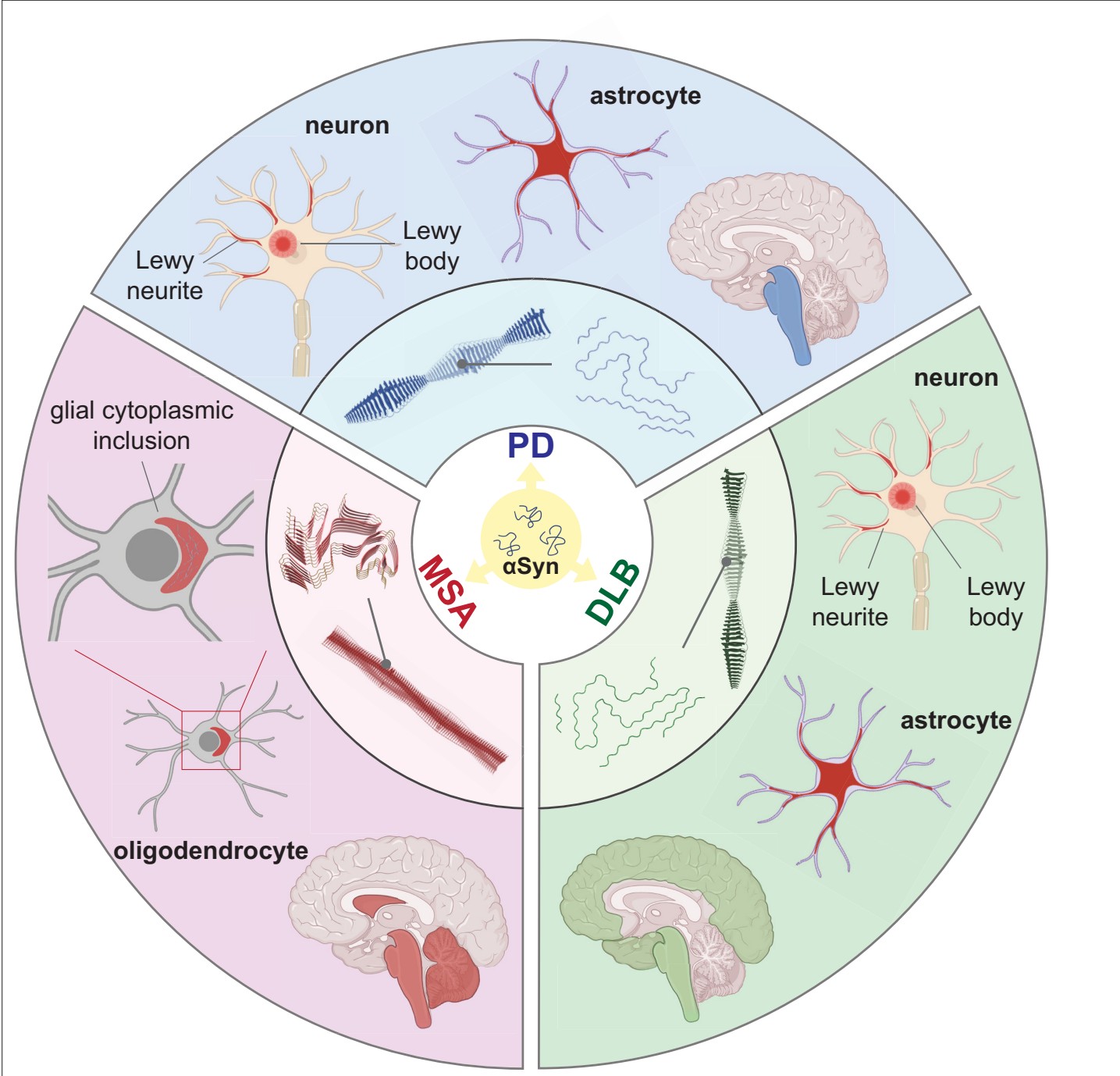

**Figure 1.** Distinct alpha-synuclein (αSyn) strains are associated with different neuropathological and clinical hallmarks of Parkinson's disease (PD), dementia with Lewy bodies (DLB), and multiple system atrophy (MSA). αSyn misfolds and aggregates into fibrils with characteristic conformations. At the atomic level, PD and DLB strains share a 'Lewy fold' structure at the fibrillar core and comprise a single protofilament (*Yang et al., 2022*). MSA strains are twisted with two protofilaments intertwined, forming a different core structure to the 'Lewy fold' (*Schweighauser et al., 2020*). At a cellular level, PD is characterized by significant neuronal loss at the brainstem, especially substantia nigra (SN), highlighted in blue. Lewy body (LB) and Lewy neurite (LN) accumulate in neurons. αSyn also accumulates in astrocytes, forming astroglial pathology. In DLB, the brainstem and neocortex are the most affected regions, highlighted in green. Here, LB and LN accumulate in neurons, and astroglial pathology is also observed. In MSA, the cerebellum, basal ganglia, and brainstem are the most affected, highlighted in red. Here, αSyn accumulates as glial cytoplasmic inclusions (GCIs) in the oligodendrocytes. The 3D structures of the αSyn fibrillar cores were generated using PyMOL, and the PDB structures from *Schweighauser et al., 2020*; *Yang et al., 2022*.

methods incorporate a key characteristic of pathological αSyn, which is its ability to induce aggregation of monomeric αSyn into different complex conformers. When pathological αSyn is present in the biosample, it seeds aggregation of the recombinant αSyn in the reaction through repeated elongation-fragmentation cycles (*Concha-Marambio et al., 2023*). A major difference between the two methods lies in the method of aggregate fragmentation (*Coysh and Mead, 2022*). RT-QuIC uses physical shaking, and PMCA uses sonication. Also, the RT-QuIC reaction is monitored automatically in real-time using a fluorophore such as Thioflavin-T (ThT), whereas PMCA requires manual measurement. Furthermore, it should be noted that the RT-QuIC products are not infectious, while the PMCA produces infectious aggregates with specific strain fidelity (*Raymond et al., 2020*). In this study, RT-QuIC has been selectively used and referred to as 'SAA'.

SAA, together with different biosamples such as cerebrospinal fluid (CSF) (*Fairfoul et al., 2016*; *Poggiolini et al., 2022*), brain homogenates (*Groveman et al., 2018*; *Sano et al., 2018*), and skin punctures (*Mammana et al., 2020*; *Kuzkina et al., 2021*), has presented its potential as a diagnostic method but also as a research tool to amplify disease-relevant fibrils. The newly amplified SAA fibrils are assumed to encode the intrinsic properties of the original seeding fibrils, thus being representative of the disease-specific strains. Therefore, using the 'model' SAA fibrils, studies have shown disease-specific structural, biochemical, and phenotypic differences, suggesting the presence of distinct conformational αSyn strains in different synucleinopathies (*Strohäker et al., 2019*; *Shahnawaz et al., 2020*; *Van der Perren et al., 2020*; *Frieg et al., 2022*).

Despite such extensive strain characterisation performed with the SAA fibrils, whether the SAA fibrils are representative models of the original seed fibrils is unclear. Studies highlighted the significant structural differences between the human brain-derived and SAA fibrils (*Lövestam et al., 2021*) and seeded pathology in vivo (*Van der Perren et al., 2020*). Nevertheless, there is currently a limited understanding of whether the resulting SAA fibrils preserve other properties, including biochemical, biophysical, cellular toxicity, and pathology.

Here, we aim to investigate the biophysical differences between (1) the brain-derived αSyn fibrils in different synucleinopathies and (2) the brain-derived αSyn versus the corresponding SAA fibrils. We examine the brain-derived αSyn fibrils extracted from the brains of patients with PD, PDD, DLB, and MSA and the brain-derived versus SAA αSyn fibrils and show striking, differences in their biochemical profile, structure, and phosphorylation pattern. Our findings reveal further evidence supporting the molecular diversity among αSyn fibrils from different synucleinopathies. Also, our study highlights the limitations of the SAA fibrils to fully mirror the brain-derived αSyn fibrils and the need to study further the mechanism of seeding amplification and the application of SAA fibrils.

## Results

### Detection of differential αSyn seeding activity from brain-derived fibrils of PD, PDD, DLB, and MSA

We extracted αSyn fibrils from the human brain of patients with PD (n=3), PDD (n=3), DLB (n=3), MSA (n=3), and healthy controls (n=3) (*Supplementary file 1*). The entorhinal cortex was used for PD, PDD, and DLB, and the striatum for MSA, as these are some of the main affected regions with αSyn pathology in each disorder. The brain-derived fibrils were diluted 1: 1000 and used as a seed in an SAA reaction to generate in vitro amplified fibrils (SAA fibrils). PD, PDD, DLB, and MSA brain-derived αSyn fibrils reached the maximum ThT fluorescence within 100 hr (*Figure 2A*). The healthy controls did not display seeding potential except for case 2 (HC 2), which showed an increasing signal towards the end of the reaction (approximately 91 hr) (*Figure 2A*). The unseeded recombinant αSyn remained negative in all reactions (*Figure 2A*). The raw data of all the replicates are presented in *Figure 2—figure supplement 1*. Kinetic parameters revealed a statistical difference in the seeding activities between PD/MSA and PDD/DLB. The time to reach 50% of the maximum fluorescence ($T_{50}$) and the lag phase (time to reach five standard deviations of the minimum fluorescence) were significantly lower in PDD/DLB than in PD/MSA (Figure fig:*Figure 2B and C*). As a result, the area under the curve (AUC) was statistically higher in PDD/DLB than in PD/MSA (*Figure 2D*). However, the maximum rate of increase in fluorescence ($V_{MAX}$) was not statistically different among the diseases (*Figure 2E*). Overall, the SAA kinetic parameters suggest PDD and DLB brain-derived αSyn fibrils have a more aggressive seeding capacity than PD and MSA.

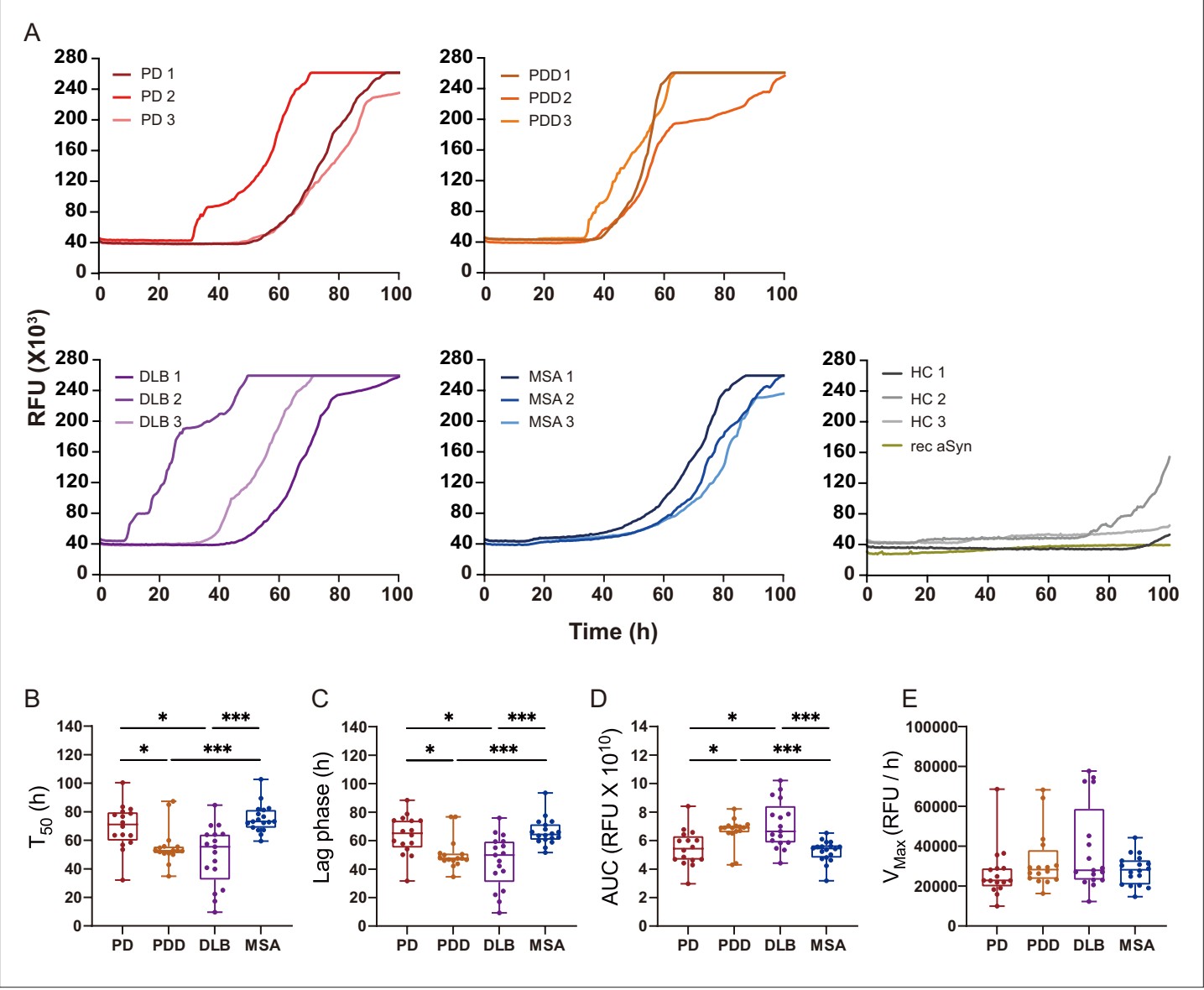

**Figure 2.** Alpha-synuclein (αSyn) seeding amplification assay (SAA) seeded with αSyn fibrils from Parkinson's disease (PD), Parkinson's disease with dementia (PDD), dementia with Lewy bodies (DLB), and multiple system atrophy (MSA) brains. (**A**) SAA was performed with sarkosyl insoluble fractions of PD (n=3), PDD (n=3), DLB (n=3), MSA (n=3), and HC (n=3) brains. The curves represent an average of six replicates. Rec αSyn indicates an unseeded control reaction. (**B**) Time to reach 50% of the maximum fluorescence ($T_{50}$). (**C**) The lag phase was taken at the time point where each positive reaction exceeded the threshold (RFU ≥ 5 SD). (**D**) Area under the curve (AUC). (**E**) The largest increase of fluorescence per unit of time ($V_{MAX}$). (**B–E**) Plotted values represent the six replicates of the three cases for each disease (n=18). RFU, relative fluorescence unit; SD, standard deviation. *p≤0.05, **p≤0.01, ***p≤0.005.

The online version of this article includes the following figure supplement(s) for figure 2:

**Figure supplement 1.** Raw data of the alpha-synuclein (αSyn) seeding amplification assay (SAA) reaction seeded with αSyn fibrils from Parkinson's disease (PD), Parkinson's disease with dementia (PDD), dementia with Lewy bodies (DLB), multiple system atrophy (MSA), and healthy control brains.

## Brain-derived and SAA αSyn fibrils display distinct biochemical profiles

At the end of the SAA reaction, the SAA αSyn fibrils were collected by ultracentrifugation. Having established that prion strains can be characterized by distinct biochemical profiles (*Lau et al., 2020*; *Shahnawaz et al., 2020*; *Sano et al., 2014*), we examined this characteristic in the brain-derived and SAA αSyn fibrils. First, the fibrils were denatured with increasing concentrations of guanidine hydrochloride (GdnHCl, 0–5 M) and then digested with proteinase-K (PK, 1 µg/ml).

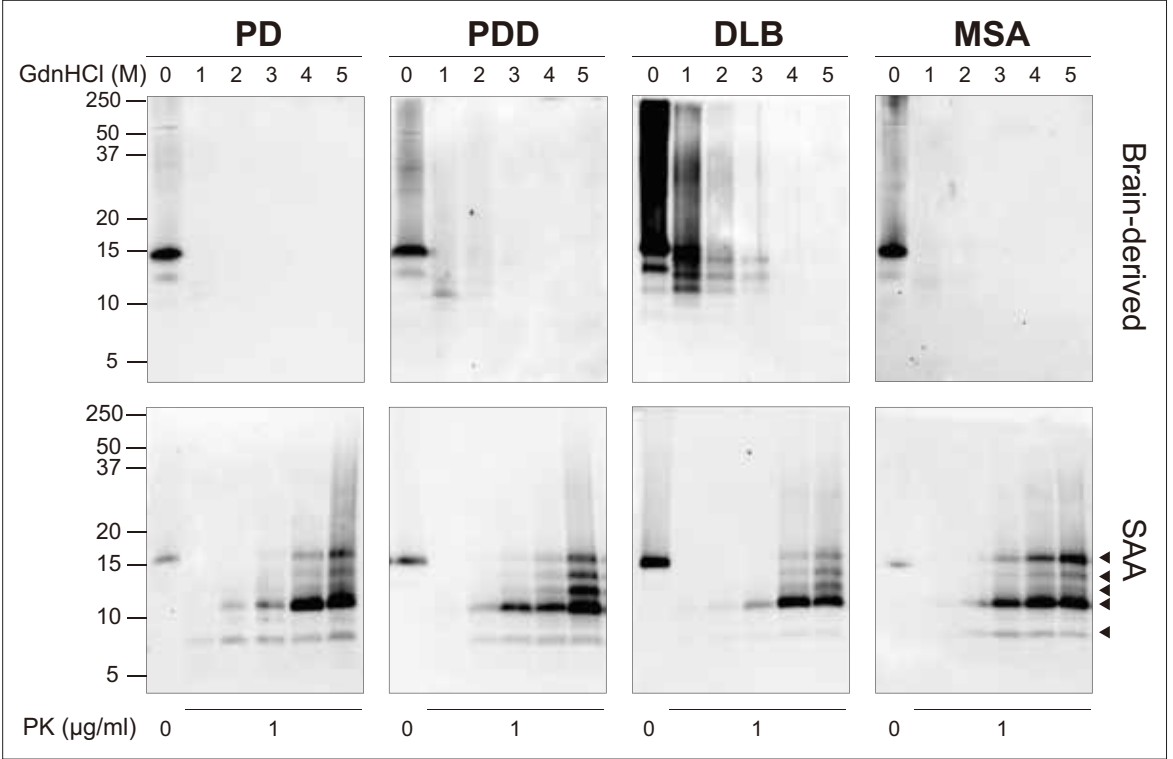

**Figure 3.** Brain-derived and SAA alpha-synuclein (αSyn) fibrils exhibited distinct biochemical profiles. Brain-derived and SAA αSyn fibrils were subjected to denaturation with increasing concentrations of GdnHCl (0–5 M) and to PK digestion (1 μg/ml). The antibody clone 42 (BD Biosciences) was used to reveal the PK-resistant peptides. Immunoblots of one representative case from Parkinson's disease (PD), Parkinson's disease with dementia (PDD), dementia with Lewy bodies (DLB), and multiple system atrophy (MSA) are presented. The brain-derived and SAA fibrils are on the top and bottom rows, respectively. The molecular weights of the protein standards are shown in kilodaltons (kDa). Black arrows mark the five PK-resistant peptides revealed in the SAA fibrils. SAA, seeding amplification assay; GndHCl, guanidine-hydrochloride; PK, proteinase-K.

The online version of this article includes the following source data and figure supplement(s) for figure 3:

**Figure supplement 1.** The full biochemical profiles of the brain-derived and SAA alpha-synuclein (αSyn) fibrils from all the cases.

**Source data 1.** Unedited western blots.

**Source data 2.** Labelled unedited western blots.

**Figure supplement 1—source data 1.** Unedited western blots.

**Figure supplement 1—source data 2.** Labelled unedited western blots.

The weakest GdnHCl (1 M) treatment completely denatured the PD and MSA brain-derived αSyn fibrils, as they were no longer visible on the immunoblot (*Figure 3*). This result was consistent among all the three cases analyzed (*Figure 3—figure supplement 1*). Higher GdnHCl concentrations of 2 M and 3 M denatured PDD and DLB brain-derived αSyn fibrils, respectively (*Figure 3*). The results illustrate that PD and MSA brain-derived αSyn fibrils have weaker biochemical stability than PDD and DLB fibrils.

The disappearance of the bands on the immunoblot after GndHCl treatment limited the comparative study of the αSyn fibrils between synucleinopathies. Therefore, we repeated the experiment using PK alone to improve the visualization of the biochemical profiles. The brain-derived αSyn fibrils were treated with five different PK concentrations starting from 0 mg/ml, incrementing to 1 mg/ml. PD cases contained mostly monomeric αSyn (15 kDa) that immediately degraded at the lowest PK concentration (1 ug/ml) (*Figure 4A*). In contrast, in PDD, DLB, and MSA cases, high molecular weight (MW) bands above 15 kDa were observed, indicative of αSyn aggregates (*Figure 4B–D*). Furthermore, lower MW bands (13, 10, 7 kDa) were also observed. As the PK concentration increased, the PDD and DLB fibrils gradually degraded, as the high MW bands started to disappear and low MW bands below 15 kDa began to appear (*Figure 4B and C*). Similarly, the MSA fibrils gradually degraded as the PK concentration increased (*Figure 4D*). However, low MW bands below 15 kDa were less prominent

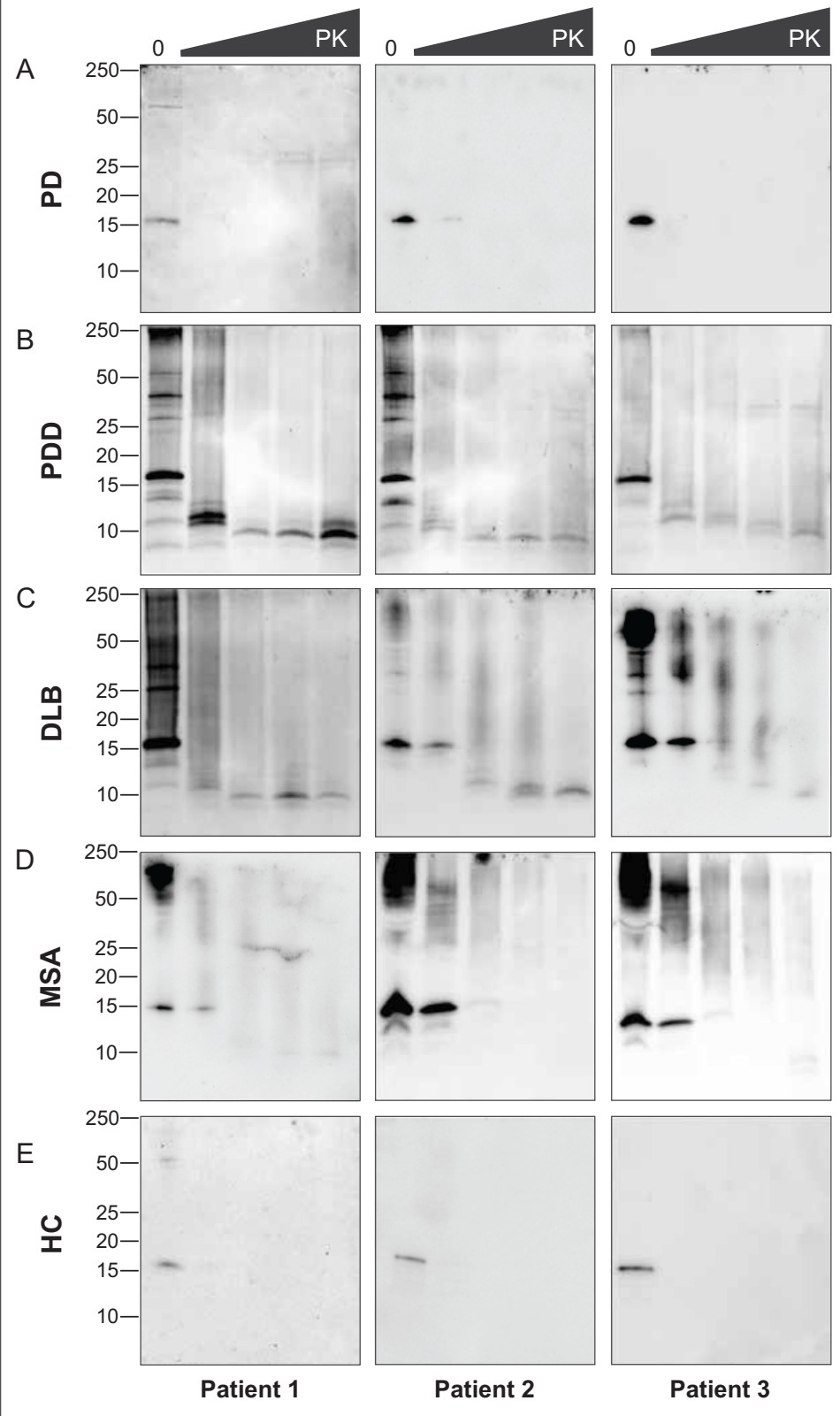

**Figure 4.** Proteinase-K degradation patterns of the brain-derived alpha-synuclein (αSyn) fibrils. The brain-derived αSyn fibrils from (**A**) Parkinson's disease (PD), (**B**) Parkinson's disease with dementia (PDD), (**C**) dementia with Lewy bodies (DLB), (**D**) multiple system atrophy (MSA), and (**E**) HC were subjected to increasing concentrations of proteinase-K (PK) at 0, 0.001, 0.01, 0.1, and 1 mg/ml, represented by the escalating triangular bar. Western blot

*Figure 4 continued on next page*

*Figure 4 continued*

was performed using the antibody clone 42 (BD Biosciences). The molecular weights of the protein standards are shown in kilodaltons (kDa).

The online version of this article includes the following source data and figure supplement(s) for figure 4:

**Figure supplement 1.** Immunohistochemical detection of alpha-synuclein (αSyn) in Parkinson's disease (PD), Parkinson's disease with dementia (PDD), dementia with Lewy bodies (DLB), and multiple system atrophy (MSA) brains.

**Source data 1.** Unedited western blots.

**Source data 2.** Labelled unedited western blots.

than those of PDD and DLB. These low MW bands were absent in HC (*Figure 4E*), and PD and HC showed no difference in the PK profiles.

Although we experimented with an identical amount of total protein (10 ug), the amount of αSyn fibril could differ in each brain-derived sample. Using immunohistochemistry, we observed a lower load of αSyn pathology in PD and MSA brains than in PDD and DLB (*Figure 4—figure supplement 1*). In addition, the brain-derived fibrils were immunogold-labeled with anti-pαSyn and negatively stained for transmission electron microscopy (TEM). The samples were imaged with TEM at low magnification to observe αSyn fibril density and distribution. PD and MSA cases contained 1–3 αSyn fibrils per region of interest (ROI), whereas PDD and DLB cases contained 3–5 αSyn fibrils per ROI (*Figure 6—figure supplement 1*). Adjacent slot blots also confirmed a higher amount of αSyn fibrils in PDD and DLB than in PD and MSA brain-derived samples. Therefore, lower fibril concentration in the PD and MSA brains could have limited the observation of the PK-resistant peptides using immunoblotting.

The SAA fibrils revealed markedly different biochemical profiles to the brain-derived αSyn fibrils. We observed identical profiles among all SAA fibrils from PD, PDD, DLB, and MSA. PK-resistant bands of 7 and 10 kDa started to appear after treatment with 2 M GdnHCl (*Figure 3*). However, with increasing concentrations of GdnHCl, specific PK-resistant peptides appeared, consisting of five bands (7, 11, 12, 13, 14 kDa) (*Figure 3*). These findings suggest that harsher denaturing conditions are required to destabilize the SAA fibrils and expose PK digestion sites. We concluded that SAA αSyn fibrils are more stable and characterized by distinct biochemical properties than the brain-derived αSyn fibrils.

## Brain-derived and SAA αSyn fibrils show structural differences

Next, we used TEM to investigate the structure of the brain-derived and SAA αSyn fibrils. The brain-derived αSyn fibrils from PD, PDD, and DLB exhibited two structures in TEM: straight and twisted (*Figure 5—figure supplement 1*). The fibrils from MSA were predominantly straight, with the rare presence of twisted type. Therefore, we only considered straight fibrils when comparing the dimensions of the fibrils among synucleinopathies. The length of the fibrils from different synucleinopathies ranged between 68–885 nm. Only the PDD fibrils were significantly longer than DLB and MSA (*Figure 5B*). The width ranged from 7 to 21 nm, and there were significant differences between the diseases. PD had the widest fibrils, followed by MSA/DLB and PDD (*Figure 5C*).

On the other hand, the SAA fibrils demonstrated a single dominant structure. PD, PDD, and DLB SAA fibrils were all straight, and MSA SAA fibrils were all twisted (*Figure 5A*). Unlike the brain-derived fibrils, the SAA fibrils were highly clustered. The length of the SAA fibrils ranged from 104 to 1008 nm, and PDD/DLB SAA fibrils were significantly longer than those of PD/MSA (*Figure 5D*). The width ranged from 8 to 25 nm, and the widths of the MSA SAA fibrils were distributed into two groups due to the alternating widths arising from the twisted structure. Unlike the brain-derived fibrils, the average widths of the SAA fibrils were similar between different synucleinopathies (*Figure 5E*).

Overall, in DLB, the SAA fibrils were significantly longer and wider than the brain-derived fibrils (*Figure 5—figure supplement 2*). SAA fibrils from PDD were wider than the brain-derived fibrils but not different in length. The dimensions of the brain-derived and SAA fibrils from PD and MSA were not significantly different. However, it was difficult to compare the dimensions of the MSA fibrils as the brain-derived and SAA fibrils had different morphologies. Our observations demonstrate that brain-derived and SAA αSyn fibrils exhibit distinct TEM structures.

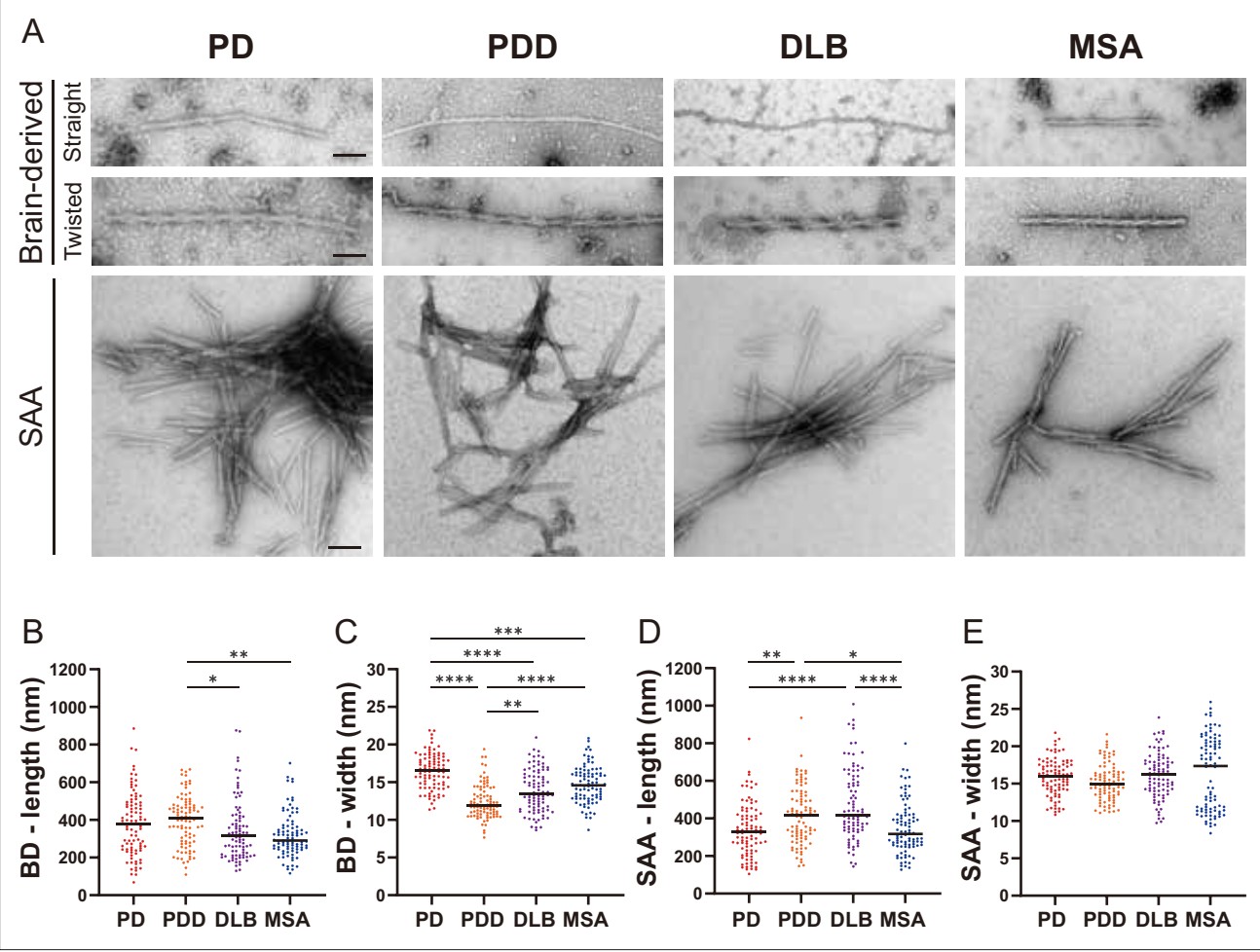

**Figure 5.** Transmission electron microscopy revealed different structures of brain-derived and seeding amplification assay (SAA) alpha-synuclein (αSyn) fibrils. (**A**) Electron microscope image of negatively stained brain-derived (BD) and SAA fibrils from Parkinson's disease (PD), Parkinson's disease with dementia (PDD), dementia with Lewy bodies (DLB), and multiple system atrophy (MSA) brains. (**B, C**) The lengths and widths of brain-derived fibrils. (**D, E**) The lengths and width of SAA fibrils. MSA SAA fibrils were twisted with alternating widths, resulting in two clusters of measurements. A total of 30 fibrils from each case were measured and plotted (n=90). Scale bar = 50 nm. *p≤0.05, **p≤0.01, ****p≤0.001.

The online version of this article includes the following figure supplement(s) for figure 5:

**Figure supplement 1.** Transmission electron microscopy (TEM) images of the twisted brain-derived alpha-synuclein (αSyn) fibrils.

**Figure supplement 2.** Comparison of the length and width of the brain-derived and SAA alpha-synuclein (αSyn) fibrils.

## Brain-derived and SAA αSyn fibrils have distinct phosphorylation patterns

To confirm the identity of the fibrils imaged by TEM, we used immunogold with αSyn fibril conformation specific (MJFR-14) and pS129-specific (anti-pαSyn) antibodies. Both brain-derived and SAA fibrils from all four synucleinopathies were probed with MJFR-14, confirming that the fibrils were αSyn fibrils (*Figure 6A*). The brain-derived fibrils were also strongly labeled with anti-pαSyn. Both antibodies, however, did not label the twisted brain-derived fibrils from PD, PDD, and DLB (*Figure 6B*). The SAA fibrils from PD and MSA were not labeled with anti-pαSyn, but PD and MSA SAA fibrils were weakly labeled (*Figure 6A*).

We performed a slot blot using identical antibodies to confirm the immunogold-labeling results. On the slot blot, DLB brain-derived αSyn fibrils showed the strongest signal with MJFR-14 and anti-pαSyn, followed by PDD, MSA, and PD (*Figure 6C*). Interestingly, the anti-pαSyn signal from the SAA fibrils was markedly different from the brain-derived fibrils. In parallel to the anti-pαSyn immunogold results (*Figure 6A*), PDD SAA fibrils showed the highest anti-pαSyn signal (*Figure 6D*). The DLB SAA

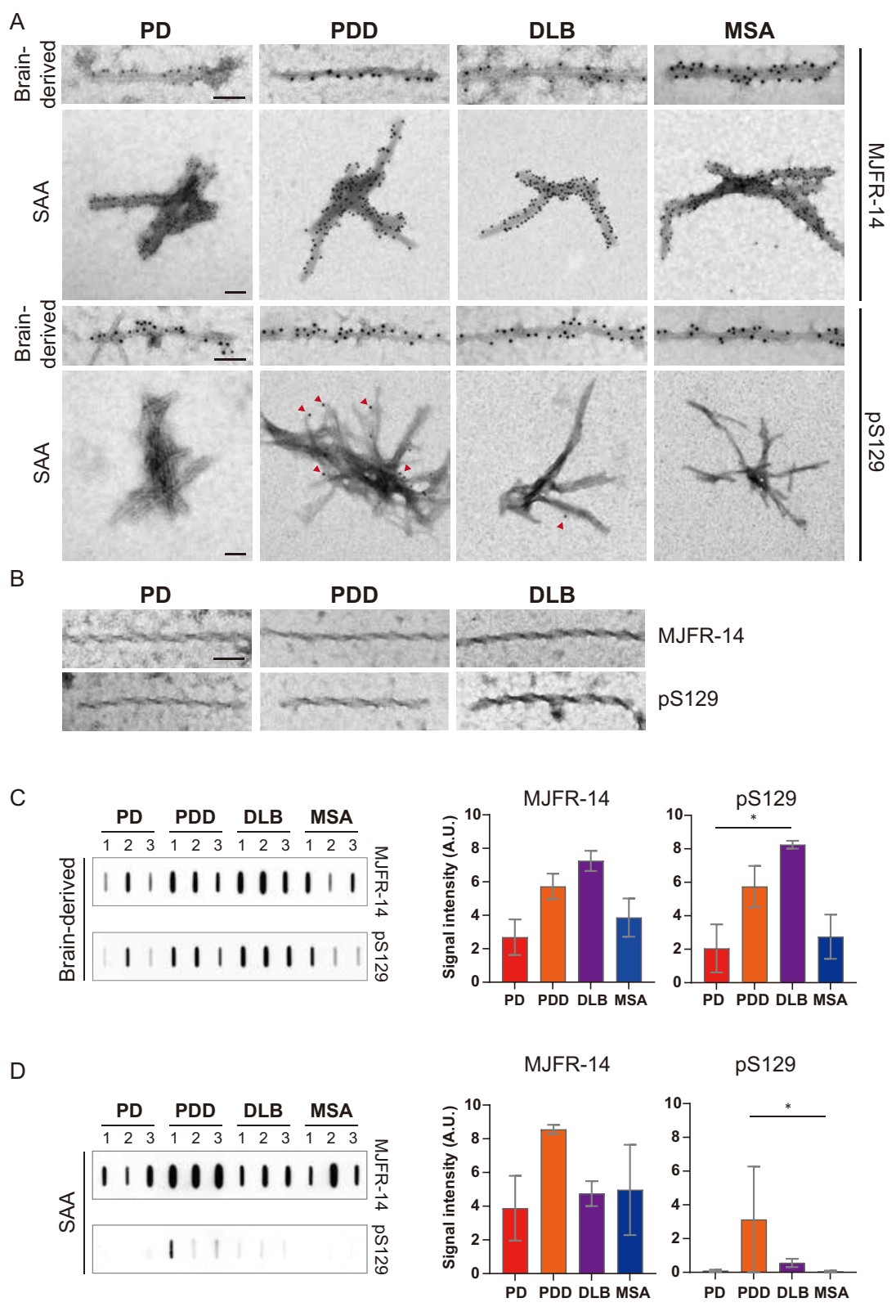

**Figure 6.** Brain-derived and seeding amplification assay (SAA) alpha-synuclein (αSyn) fibrils showed distinct phosphorylation patterns. (**A**) Electron microscope image of brain-derived and SAA αSyn fibrils from Parkinson's disease (PD), Parkinson's disease with dementia (PDD), dementia with Lewy bodies (DLB), and multiple system atrophy (MSA) brains labeled with fibril conformation-specific (MJFR-14) and anti-pαSyn (pS129) antibodies. (**B**) The twisted αSyn fibrils from PD, PDD, and DLB brains were not labeled for MJFR-14 and pS129. Scale bar = 50 nm. (**C,D**) Semi-quantification of αSyn fibrils

*Figure 6 continued on next page*

*Figure 6 continued*

and pαSyn confirm the different patterns of αSyn phosphorylation between brain-derived and SAA αSyn fibrils. The amount of αSyn fibrils and pαSyn in (**C**) brain-derived and (**D**) SAA fibrils was determined using a slot blot. Two µg of proteins were filtered on a nitrocellulose membrane and probed with MJFR-14 and pS129 antibodies. The semi-quantitative measurement was done by averaging the measurements of three cases from each disease. Error bars indicate ± one standard deviation (SD). *p≤0.05.

The online version of this article includes the following source data and figure supplement(s) for figure 6:

**Source data 1.** Unedited slot blots.

**Source data 2.** Labelled unedited slot blots.

**Figure supplement 1.** pS129 immunogold transmission electron microscopy (TEM) images of the brain-derived alpha-synuclein (αSyn) fibrils.

**Figure supplement 1—source data 1.** Unedited slot blots.

**Figure supplement 1—source data 2.** Labelled unedited slot blots.

---

fibrils were also weakly phosphorylated, while those of PD and MSA were negative. Considering the absence of phosphorylated recombinant monomers in our SAA reaction, the phosphorylated αSyn must have transmitted from the seed brain-derived fibrils. Our results demonstrate that the brain-derived and SAA αSyn fibrils display different patterns of S129 phosphorylation.

## Discussion
### Strain-like properties of the brain-derived αSyn fibrils from different synucleinopathies

Findings in this study provide evidence of distinct 'strain-like' properties of the brain-derived αSyn fibrils from different synucleinopathies. First, faster seeding kinetics were observed in the brain-derived αSyn fibrils from PDD and DLB than those from PD and MSA, suggesting a distinct 'strain' characterized by an aggressive seeding capacity. In contrast, those from PD and MSA are discrete 'strains' with milder seeding capacity.

When analysing the seeding kinetics, the different disease durations of our selected cases must be noted. In particular, our PDD cases had longer disease duration than PD, although both shared the same Braak stage 6. The Braak staging, however, only represents the distribution of LB pathology and fails to consider the load of LBs. We have shown that PDD brains showed higher LB load in the entorhinal cortex compared to PD brains. Thus, a higher accumulation of fibrils in PDD brains might have resulted in faster seeding kinetics. Nevertheless, our DLB cases had shorter disease duration than PDD, but the comparable burden of LBs and both diseases demonstrated similar seeding kinetics. Thus, the differential seeding kinetics is likely to result from strain properties or burden of pathology rather than disease duration.

Previous studies of αSyn SAA with CSF showed that PD and MSA could be distinguished using $T_{50}$, $V_{MAX}$ (**Poggiolini et al., 2022**), and $F_{MAX}$ (**Shahnawaz et al., 2020**; **Shahnawaz et al., 2017**). However, in our αSyn SAA with brain-derived fibrils, the same parameters cannot distinguish between PD and MSA. Differences in the findings are likely to arise from the type of seeds. An unprocessed CSF would likely comprise distinct and different concentrations of αSyn species. Moreover, the choice of biosample (CSF, olfactory mucosa, skin) significantly affects the outcome of seeding kinetics (**Chahine et al., 2020**). By isolating the brain-derived αSyn fibrils, we could study the seeding kinetics of the fibrils themselves and, as a result, better understand their distinct intrinsic seeding properties.

Interestingly, one healthy control case (HC 2) showed amplification towards the end of the SAA reaction. The subject had Braak αSyn stage 0, suggesting the absence of incidental Lewy bodies. Previously, positive seeding activities have been detected in HCs (**Han et al., 2020**). The exact cause of this observation is unclear, but the long reaction time might have contributed to the aggregation. SAA studies have shown that their HCs remain negative at 48–60 hr (**Groveman et al., 2018**; **Bargar et al., 2021**). Our HCs were negative at these times and only started to aggregate from 80 hr.

Next, examining the phosphorylated brain-derived αSyn fibrils (pαSyn), we demonstrate a possible correlation between the amount of pαSyn and rapid seeding kinetics. Studies have shown that pαSyn enhances αSyn aggregation in vitro (SH-SY5Y cells) and in vivo (rodent models) (**Smith et al., 2005**; **Karampetsou et al., 2017**). However, contradicting findings have also been reported, suggesting a neuroprotective role of pαSyn (**Ma et al., 2016**; **Ghanem et al., 2022**). Although the biological role

of pαSyn is unclear, we report that a higher amount of pαSyn may contribute to the more aggressive seeding kinetics observed in PDD and DLB brain-derived αSyn fibrils than those of PD and MSA.

Further evidence of 'strain-like' property was exhibited by the distinct biochemical profiles of the brain-derived αSyn fibrils from different synucleinopathies. The brain-derived αSyn fibrils from PD and MSA were more susceptible to GndHCl denaturation than PDD and DLB, indicating conformational instability of PD and MSA brain-derived αSyn fibrils. Nevertheless, it must be noted that the rapid denaturation of PD fibrils might be due to a low concentration of αSyn fibrils, as demonstrated by a lower Lewy pathology load in the entorhinal cortex. A previous study also showed that human brain-derived αSyn fibrils from DLB were more stable than MSA by using GdnHCl treatment (*Lau et al., 2020*). Interestingly, the less stable MSA fibrils propagated faster in TgM83 mice than the more stable DLB fibrils. Such inverse correlation between conformational stability and disease propagation has been characterized in prion disease (*Legname et al., 2006*). Therefore, our findings on the differential conformational stability could reflect the strain-specific characteristics of αSyn fibrils. Subsequently, it could be related to the fast disease progression of MSA compared to PDD and DLB.

Subsequently, by assessing the PK-resistance of the αSyn fibrils, we illustrate further the molecular diversity of αSyn fibrils. The sizes of the low MW PK-resistant fragments differed between diseases and cases within each disease. These distinct PK-resistant peptides suggest the presence of conformational variants indicative of αSyn fibrillar strains within a single disease entity. Similarly, Guo and colleagues reported evidence of distinct αSyn fibrillar conformers in different PDD patients' brains (*Guo et al., 2013*). Our group has also shown different SAA kinetics in the CSF samples of PD patients divided into four different subtypes based on their baseline and progressive motor and non-motor symptoms (*Lawton et al., 2018*), suggesting αSyn strain heterogeneity within the same disease (*Poggiolini et al., 2022*). Finally, unlike PDD and DLB, MSA αSyn fibrils gradually disappeared upon PK treatment without being digested into small fragments. The lack of PK-resistant peptides in MSA indicate a readily digestible property of MSA αSyn fibrils. Together with its conformational instability, these biochemical properties might indicate a specific αSyn strain in MSA.

Interestingly, MSA fibrils also showed distinct structural features compared to those from PD, PDD, and DLB. The fibrils from MSA were predominantly 'straight', and those from PD, PDD, and DLB were 'straight' and 'twisted.' While the 'straight' brain-derived fibrils were densely decorated with MJFR-14 and anti-pαSyn antibodies, confirming their identity as a phosphorylated αSyn fibril, the 'twisted' fibrils were not recognised by these proteins. Similarly, Spillantini and colleagues also reported a finding of 'twisted' fibril from the cingulate cortex of a DLB brain with no labeling by the αSyn-specific antibody (*Spillantini et al., 1998*). Further ultrastructure analysis is essential to reveals the identity of the unlabeled twisted fibrils.

The methodological limitations of the structural characterisation in this study should be considered. Cryo-electron microscope (EM) ultrastructure revealed the twisted structures of the MSA brain-derived αSyn fibrils and single twisted protofilament structures in PD, PDD, and DLB brains (*Yang et al., 2022*). Our TEM structures may differ due to the selection of different brain regions compared to those used in the literature and the limitation of the resolving power of TEM. Future cryo-EM structural studies of αSyn fibrils from not only different synucleinopathies but also different brain regions and subtypes of PD are required. Accumulating evidence already suggests the presence of distinct biochemical properties among clinical and genetic subtypes of PD (*Poggiolini et al., 2022*; *Siderowf et al., 2023*). Thus, it is essential to continue investigating the molecular profiles to understand the diversity of αSyn conformers and their role in pathogenesis.

## Distinct biophysical properties between brain-derived and SAA fibrils

In order to assess whether the SAA fibrils truly replicate the brain-derived fibrils (the seed), we compared the biophysical characteristics of the brain-derived αSyn fibrils and their respective SAA fibrils. First, by examining the resistance to GdnHCl and PK digestion, we illustrated a prominent biochemical difference between the brain-derived and SAA fibrils. High concentrations (4–5 M) of GdnHCl completely denatured the brain-derived fibrils, whilst SAA fibrils were still resistant. This indicates that the SAA fibrils are more stable and PK-resistant than their respective brain-derived fibrils. Furthermore, the SAA fibrils from all four synucleinopathies had identical patterns of PK resistance. The SAA amplifies fibrils with uniform biochemical features, losing the intrinsic properties of the original seed fibril and failing to replicate the biochemical properties of the original brain-derived fibrils.

The differences in biochemical stability might arise from the distinct distribution of the fibrils. SAA fibrils were clustered in negative-stain TEM, whereas the brain-derived fibrils were distributed in single filaments (*Figure 5*). Also, this study used a PIPES-based buffer to amplify the SAA fibrils (*Poggiolini et al., 2022*). As the microenvironmental context of αSyn amplification substantially affects the newly amplified fibrils (*Peng et al., 2018*; *Gustavsson et al., 2021*), the artificial in vitro environment can favor the formation of specific three-dimensional structures that are more conformationally stable.

Next, we showed that brain-derived fibrils display a mixture of straight and twisted structures, while SAA fibrils have a unified morphology: straight (PD, PDD, DLB) or twisted (MSA). This was an unexpected finding, as the conformational stability patterns were identical across all SAA fibrils. Here, technical limitations exist where the conformational stability was assessed using a single antibody targeting the 91–99 region of αSyn. A future study could employ multiple antibodies targeting the N- and C-terminus to further understand the relations between conformational stability and fibril morphology.

The striking structural difference may be due to a dominant seeding fibril within the mixture of brain-derived fibrils. Interestingly, straight fibrils were dominant in MSA brains, but the corresponding SAA fibrils were twisted. The commonality of a particular structure in a fibrillar mixture may not be a determinant of a dominant seeding fibril. Other factors might contribute, such as metal ions (*Uversky et al., 2001*) and post-translational modifications (PTMs), which also affect seeding efficiency (*Balana et al., 2023*). Also, our PIPES-based SAA reaction buffer with 150 mM NaCl is likely to have influenced the SAA fibril structure. However, the discrete structure of the MSA SAA fibrils compared to those of PD, PDD, and DLB implies that a complex matrix of intrinsic factors within the brain-derived fraction impacts seeding amplification and the resulting fibrillar structure.

A recent cryo-EM analysis of the MSA brain-derived and SAA fibrils revealed striking structural differences (*Schweighauser et al., 2020*; *Lövestam et al., 2021*). A few significant differences included the protofilament fold, the inter-protofilament interface, and the geometry of specific residues that shape the hydrophobic core. As the TEM has restricted resolution, future cryo-EM analyses are imperative to connect the structure to the disease heterogeneity and progression.

Lastly, we investigated the differences in the phosphorylation level between the brain-derived and SAA αSyn fibrils to identify differences in the PTM patterns. The brain-derived fibrils generally showed a higher phosphorylation level than the SAA fibrils. PDD and DLB SAA fibrils were weakly phosphorylated and decorated with pαSyn immunogold labels at the rare end of the fibrils (*Figure 6*). Combining these findings, we can argue the possibility that the SAA fibrils extended from pαSyn species dissociated from the brain-derived fibrils. Therefore, the amount of initial pαSyn fibrils in the seed might be a critical determinant of the phosphorylation state of the resulting SAA fibrils.

The dissimilarity observed in the brain-derived and SAA fibrils might result from the RT-QuIC methodology. For instance, prion studies reported that RT-QuIC end-products are non-infectious (*Coysh and Mead, 2022*). On the other hand, PMCA prion end-products are infectious in vitro and in vivo and maintain strain-specific properties (*Castilla et al., 2005*). This indicates the limitation of RT-QuIC in reproducing the toxicity of the source seeds as it loses biological information. Therefore, the biophysical differences between the brain-derived and SAA fibrils might result from the limitations of the RT-QuIC methodology.

Studies have explored the pathogenicity of brain-derived αSyn fibrils in vitro using primary neurons, oligodendrocytes, and HEK cell lines expressing mutated αSyn (*Woerman et al., 2018a*; *Peng et al., 2018*; *Woerman et al., 2015*; *Yamasaki et al., 2019*). Moreover, similar pathological characterization has been done in different mouse models (*Lau et al., 2020*; *Woerman et al., 2019*; *Woerman et al., 2018a*; *Prusiner et al., 2015*; *Peng et al., 2018*; *Masuda-Suzukake et al., 2013*; *Watts et al., 2013*; *Bernis et al., 2015*; *Woerman et al., 2018b*; *Woerman et al., 2020*). Interestingly, both in vitro and in vivo, MSA brain-derived αSyn induced significant αSyn aggregation, whereas those from PD/PDD and DLB brains produced either no or low αSyn accumulation. Similar disease-specific patterns were also observed in SAA-amplified fibrils (*Shahnawaz et al., 2020*; *Van der Perren et al., 2020*; *Tanudjojo et al., 2021*). In future studies, it would be crucial to examine the brain-derived and SAA fibrils in comparison in cellular and mouse models for their pathological characteristics.

In summary, the distinct biochemical profiles and structures of the brain-derived αSyn fibrils from different synucleinopathies provide supporting evidence of molecular diversity of brain-derived αSyn fibrils among different synucleinopathies and individual patients. Moreover, the SAA fibrils failed to

adopt the biochemical, structural, and PTM properties of the seed brain-derived αSyn fibrils, which addresses a significant limitation of SAA in replicating the intrinsic biophysical properties of the seed fibrils. Therefore, our finding highlights the necessity of re-evaluating the SAA seeding mechanism and its capacity to generate disease-relevant αSyn fibrils used in different in vitro and in vivo models. Furthermore, it remains essential to investigate the human-derived αSyn fibrils in the different brain regions and in other peripheral sites to explore αSyn strains and how they may affect the progression of pathology. They also remain the most disease-specific fibrils used in cellular and animal models.

## Materials and methods

### Patient collection

Brain tissues from patients with PD and PDD were obtained from the Parkinson's UK Brain Bank (Imperial College London, UK) in accordance with approved protocols by the London Multicentre Research Ethics Committee. Brain tissues from patients with DLB, MSA, and HC subjects were collected from the Oxford Brain Bank (OBB, University of Oxford, UK) in accordance with approved protocols by the South Central - Oxford C Research Ethics Committee (ref 23/SC/0241). All participants had given prior written informed consent for the brain donation. Both brain banks comply with the requirements of the Human Tissue Act 2004 and the Codes of Practice set by the Human Tissue Authority (HTA licence numbers 12275 for the Imperial and 12217 for OBB). The clinico-pathological demographics of the patients and healthy controls are summarized in Table tab:*Supplementary file 1*.

### Extraction of αSyn fibrils from the human brain

Sarkosyl insoluble αSyn fibril was extracted from the brains of PD (n=3), PDD (n=3), DLB (n=3), MSA (n=3) patients, and healthy controls (n=3). The entorhinal cortex was selected for PD, PDD, and DLB, and the striatum was selected for MSA. The extraction protocol was adapted from the method presented by Schweighauser and colleagues (*Schweighauser et al., 2020*). Brain tissues (0.5 g) were homogenized in an extraction buffer of 10 mM Tris-HCl, 0.8 M NaCl, 10% sucrose, and 1 mM EGTA (pH 7.5). Sarkosyl was added to a final concentration of 2% and incubated for 30 min at 37°C. Homogenates were centrifuged at 10,000 g for 10 min at 4°C. The supernatants were centrifuged at 100,000 g for 60 min at 4°C. The pellet was washed, resuspended in 500 µl/g of extraction buffer, and centrifuged at 500 g for 1 min at 4°C. The resulting supernatants were diluted 1/3 in buffer consisting of 50 mM Tris-HCl, 0.15 M NaCl, 10% sucrose, and 0.2% sarkosyl (pH 7.5). The diluted supernatant was centrifuged at 100,000 g for 60 min at 4°C. The resulting pellet was washed and resuspended in 250 µl/g of 30 mM Tris-HCl (pH 7.5), being the sarkosyl-insoluble fraction. The protein concentration was determined using the bicinchoninic acid assay (BCA) (Thermo Scientific). The sarkosyl insoluble fraction was aliquoted and stored at –80°C.

### αSyn SAA

The brain-derived αSyn fibrils were diluted 1:1000 in 30 mM Tris-HCl (pH 7.5), and 2 µl was added to 98 µl of the reaction mixture consisting of 100 mM PIPES (pH 7), 150 mM NaCl, 0.1 mg/ml of recombinant αSyn (rPeptide) and 10 µM thioflavin-T (ThT) to a final reaction volume of 100 µl. Before use, the recombinant αSyn was filtered through a 100 kDa molecular weight cut-off (MWCO) filter. The reaction mixture was loaded onto a black 96-well plate with a clear bottom (Nalgene Nunc). The plate was sealed with a plate sealing film (Thermo Scientific) and incubated at 42°C in a BMG FLUOstar Omega plate reader for 100 hr with cycles of 1 min shaking (400 rpm double orbital) and 1 min rest. ThT fluorescence measurement (450 nm excitation and 480 nm emission) was taken every 30 min. Each sample was run in six replicates. The reaction end-products were ultracentrifuged at 100,000 g for 1 hr at 4°C and collected as SAA fibril.

### Conformational stability assay and immunoblotting

The brain-derived αSyn fibrils (10 µg) and SAA fibrils (1 µg) were treated with different GdnHCl concentrations (0–5 M) at 37°C for 1 hr on a thermoshaker (800 rpm). The reactions were stopped by reducing the GdnHCl concentration to 0.5 M. The fibrils were treated with 1 µg/ml of proteinase-K (PK) at 37°C for 30 min on a thermoshaker (500 rpm). Digested samples were collected using ultracentrifugation at 100,000 g, 4°C for 1 hr. Tricine sample buffer (Bio-Rad) was added to the samples and

boiled at 95°C for 10 min. The samples were analyzed with 16.5% Tris-Tricine gels (Bio-Rad) and immunoblotted on nitrocellulose membranes (Amersham). Membranes were blocked with 5% skimmed milk in TBS-Tween and incubated with anti-αSyn clone 42 (BD Biosciences, 1:1000 dilution). In a slot blot, 2 µg of protein was immobilized on a nitrocellulose membrane (Amersham) by filtration using a slot blot apparatus (GE Healthcare). The membrane was blocked with 5% skimmed milk and probed with MJFR-14-6-4-2 (Abcam, 1:3000 dilution) and EP1536Y (Abcam, 1:1,000 dilution). The membranes were developed using an ECL western blot detection kit (Amersham). Semi-quantitative analysis of the slot blot data was performed using ImageJ.

## TEM and immunolabeling electron microscopy

Five µl of 0.5 µg brain-derived αSyn fibrils and 5 µl of SAA fibrils were applied to glow-discharged carbon grids (Agar Scientific, 300 mesh) and incubated for 2 min. The grid was washed with water for 10 s and negatively stained with 2% uranyl acetate for 10 s. Stained samples were imaged on an FEI Tecnai T12 microscope operated at 120 kV.

For immunolabeling, the fibrils were applied to glow-discharged carbon grids and blocked with a blocking buffer (0.2% fish gelatin in PBS). Then, the grid was incubated with antibodies targeting the phosphorylated αSyn (EP1536Y, Abcam, 1:20 dilution) and αSyn fibril conformer (MJFR-14-6-4-2, Abcam, 1:50 dilution). After washing with blocking buffer, the sample was incubated with goat anti-rabbit IgG coupled with 10 nm gold (Ab27234, Abcam) diluted 1:10 in the blocking buffer. Then, the grid was fixed with 0.1% glutaraldehyde in PBS. The samples were stained with 2% uranyl acetate for 30 s.

## Immunohistochemistry

Six µm-thick sections of formalin-fixed paraffin-embedded (FFPE) human brain tissue sections were de-paraffinized in xylene (3x5 min) and rehydrated through decreasing concentration of industrial denatured alcohol (IDA) (100%, 100%, 90%, 70%; 5 min each) and subsequently in distilled water (5 min). For antigen retrieval, sections were treated with 70% formic acid for 15 min. The sections were then blocked with 3% $H_2O_2$ for 20 min and with 10%fetal bovine serum (FBS) in PBS for 30 min. Sections were incubated with anti-αSyn clone 42 (BD Biosciences) diluted 1:1000 in 10%FBS overnight at 4°C. After washing with PBS (3×5 min), sections were incubated with Ms/Rb-HRP secondary antibody (Agilent Technologies) for 30 min at RT. Sections were developed with Dako DAB (Agilent Technologies) and counterstained with haematoxylin. Finally, sections were mounted to coverslips with DPX (Thermo Scientific).

## Statistical analysis

Mann-Whitney U test was used when comparing two independent groups. Independent Kruskal-Wallis Test was performed to compare more than three groups for SAA kinetic parameters *Figure 2*, fibril dimensions (PD, PDD, DLB, and MSA) *Figure 5*, and the amount of phosphorylated αSyn fibrils *Figure 6*. p-values <0.05 were considered statistically significant. All statistical analyses were performed using SPSS Statistics, Version 28.

## Acknowledgements

We thank all the patients and their families for the valuable brain donations for research. We also want to thank Dr. Errin Johnson and the Oxford bioimaging facility for providing access and training for transmission electron microscopy and assisting with acquiring the imaging data. We acknowledge Dr. Alan King Lun Liu for helping us with the case selection and analysis of the clinical summary of the patients. We acknowledge the Oxford Brain Bank, supported by Brains for Dementia Research (BDR) (Alzheimer Society and Alzheimer Research UK) and the National Institute for Health Research (NIHR) Oxford Biomedical Research Centre (BRC). LP is supported by the GSK-Institute of Molecular & Computational Medicine, NIHR Oxford BRC, Parkinson Foundation, the Michael J Fox Foundation, the Galen and Hilary Weston Foundation, and the National Institute of Health. LC is supported by NIHR Oxford BRC.

# Additional information

## Funding

| Funder | Grant reference number | Author |
|---|---|---|
| Oxford-GSK Institute of Molecular and Computational Medicine | | Laura Parkkinen |
| NIHR Oxford Biomedical Research Centre | | Livia Civitelli<br>Laura Parkkinen |
| Parkinson's Foundation | | Laura Parkkinen |
| Michael J Fox Foundation for Parkinson's Disease Research | | Laura Parkkinen |
| Galen and Hilary Weston Foundation | | Laura Parkkinen |
| National Institutes of Health | | Laura Parkkinen |

The funders had no role in study design, data collection and interpretation, or the decision to submit the work for publication.

## Author contributions

Selene Seoyun Lee, Conceptualization, Data curation, Formal analysis, Investigation, Methodology, Writing - original draft, Project administration, Writing – review and editing; Livia Civitelli, Conceptualization, Data curation, Formal analysis, Supervision, Investigation, Methodology, Writing – review and editing; Laura Parkkinen, Conceptualization, Supervision, Funding acquisition, Writing – review and editing

## Author ORCIDs

Selene Seoyun Lee ⓘ https://orcid.org/0000-0002-2801-6202
Livia Civitelli ⓘ http://orcid.org/0000-0003-2399-1436
Laura Parkkinen ⓘ https://orcid.org/0000-0002-3392-8564

## Ethics

Human subjects: Brain tissues from patients with PD and PDD were obtained from the Parkinson's UK Brain Bank (Imperial College London, UK) in accordance with approved protocols by the London Multicentre Research Ethics Committee. Brain tissues from patients with DLB, MSA and healthy controls were collected from the Oxford Brain Bank (OBB, University of Oxford, UK) in accordance with approved protocols by the South Central - Oxford C Research Ethics Committee (ref 23/SC/0241). All participants had given prior written informed consent for the brain donation. Both brain banks comply with the requirements of the Human Tissue Act 2004 and the Codes of Practice set by the Human Tissue Authority (HTA licence numbers 12275 for the Imperial and 12217 for OBB).

Reviewer #2 (Public review): https://doi.org/10.7554/eLife.92775.3.sa1
Author response https://doi.org/10.7554/eLife.92775.3.sa2

# Additional files

## Supplementary files

• Supplementary file 1. Clinicopathological summary of the patients included in this study. The table summarizes the clinical and pathological reports of the patients included in this study. PD, Parkinson's disease; PDD, Parkinson's disease with dementia; DLB, dementia with Lewy body; MSA, multiple system atrophy; HC, healthy control; ECtx, entorhinal cortex; BG, basal ganglia; M, male; F, female; AD, Alzheimer's disease; CVD, cardiovascular disease; and alpha-synuclein (αSyn).

• MDAR checklist

## Data availability

The study did not generate any novel datasets.

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

## Appendix 1

### Cases

The clinical diagnosis of Parkinson's disease patients was performed using the UK Parkinson's Disease Brain Bank criteria, followed by a neurologist review (*Daniel and Lees, 1993*). PDD patients were diagnosed using the Movement Disorder Society task force PDD diagnostic criteria (*Dubois et al., 2007*). DLB patients were diagnosed using the DLB consortium diagnostic criteria (*McKeith et al., 2017*). MSA patients were diagnosed based on the MSA consensus statement (*Gilman et al., 1998*). All PD, PDD, and DLB subjects had neuropathologically confirmed Lewy body pathology. Neuropathological diagnosis of MSA subjects was based on degeneration of striatonigral and olivopontocerebellar regions combined with glial cytoplasmic aSyn inclusions (*Papp et al., 1989*). Braak αSyn and tau stages were assessed according to the protocol outlined by BrainNet Europe (*Alafuzoff et al., 2008*; *Alafuzoff et al., 2009*). Specifically, Braak αSyn stages were assigned by examining the distribution of αSyn inclusions in the medulla, pons, midbrain, basal ganglia, hippocampus, cingulate gyrus, temporal, frontal and parietal cortices. Braak tau stages were assigned based on tau immunostains across regions of the visual cortex, the middle temporal gyrus, the anterior hippocampus and the posterior hippocampus.

