## [Editor Report · eLife Assessment]

This **important** work compares the strain properties of a-synuclein fibrils isolated from LBD and MSA patient samples with the resulting amplified fibrils following SAA. Using orthogonal biochemical and structural approaches to strengthen their analyses, the authors provide **solid** evidence that the SAA-amplified fibrils do not recapitulate the disease-relevant strains present in the patient samples. CryoEM would further strengthen this data but it is outside the scope of the work. This work should be considered in the widespread applications of SAA in synucleopathies and its potential limitations.

---

## [Referee Report · Reviewer #2 (Public review)]

Most neurodegenerative diseases are characterized by the self-templated misfolding of a particular protein in a manner that enables progressive spread throughout the central nervous system. In diseases including Parkinson's disease (PD) and multiple system atrophy (MSA), the protein a-synuclein misfolds into unique strains, which use this self-replicating mechanism to encode disease-specific information. Previous research suggests that a major contributor to the lack of successful clinical trials across neurodegenerative diseases is the lack of disease-relevant strains used in preclinical testing. While MSA patient samples are known to replicate efficiently in cell and mouse models of disease, Lewy body disease (LBD) patient samples do not. To overcome this obstacle, the seeding amplification assay (SAA) uses recombinant a-synuclein to amplify the misfolded protein structure present in a human patient sample. The resulting fibrils are then widely used by many laboratories as a model of PD. In this manuscript, Lee et al., set out to compare the strain properties of a-synuclein fibrils isolated from LBD and MSA patient samples with the resulting amplified fibrils following SAA. Using orthogonal biochemical and structural approaches to strengthen their analyses, the authors report that the SAA-amplified fibrils do not recapitulate the disease-relevant strains present in the patient samples. Moreover, their data suggest that regardless of which strain is used to seed the SAA reaction, the same strain is generated. These results clearly demonstrate that the SAA-amplified material is likely not disease-relevant. SAA fibrils are broadly used throughout academic and pharmaceutical laboratories. They are used in ongoing drug discovery efforts and recombinant fibrils broadly inform much of what is known about a-synuclein strain biology in LBD patients. The implications of the reported work are, therefore, expansive. These findings add to the growing ledger of reasons that the use of SAA fibrils in research should be halted until improved methods for amplification with high fidelity are developed.

---

## [Author Response]

The following is the authors’ response to the original reviews.

**Reviewer 1 (Public Review):**
Comment 1. Clinical Data on Patient Brain Samples: The inclusion of specific details such as postmortem intervals and the age at disease onset for patient brain samples would be valuable. These factors could significantly affect the quality of the tissues and their relevance to the study. Moreover, given the large variation in disease duration between PD and PDD, it’s important to consider disease duration as a potential confounding factor, especially when concluding that PDD patients have a more severe form of synucleinopathy compared to PD.

We thank the reviewer for this valuable comment. We have included the post-mortem interval (PMI) and age of death in Table S1, showing the clinicopathological information. Changes on page 16. As suggested by the reviewer, we included the discussion on the large variation in disease duration between PD and PDD cases. We noted that DLB cases also have shorter disease durations but still demonstrate seeding kinetics similar to PDD. Therefore, we hypothesise that the molecular differences we observed between different diseases were due to the strain properties or higher pathological load (seen in both PDD and DLB) and are unlikely due to the disease duration. Changes on pages 9-11, lines 204-212.

Comment 2. Inclusion of Healthy Controls in Multiple Tests: Given the importance of healthy controls in scientific studies, especially those involving human brain samples, the authors could consider using healthy controls in more tests to strengthen the robustness of the findings. Expanding the use of healthy controls in biochemical profiling and phosphorylation profiles would provide a better basis for comparison and clarify the significance of results in a disease context. This will help the authors to elaborate on the interpretation of results, for example, in Figure 3, where the authors claim that PD brains show mostly monomeric _α*Syn forms (line 119 and 120, and also in 222 and 223). Whether it implies the absence of alpha-syn pathology in PD brains? If there are differences from healthy controls? What are these low molecular weight bands (¡15kD) (line 125-126) and whether they are also present in healthy controls? Also, we do not have a perfect pS129-specific (anti-p*α_Syn) antibody. They are known for non-specific labeling. Investigating the phosphorylation levels in healthy controls and comparing them to PD brains, especially considering the predominance of monomeric (healthy _α_Syn?) in PD brains, would help clarify the observed changes.

We agree with the reviewer’s assessment and consider this an important suggestion. We performed biochemical profiling and immunogold imaging with the three HC cases and presented the results in Figure 4. aSyn in healthy controls was completely digested by PK. The low MW bands were absent in PD and HC, and there was no difference in the PK profiles. However, this may be due to the low pathology load and amount of pathological aSyn in the selected PD brains. Additional comments were added to the results. Changes are on pages 4 (lines 136-137) and page 7 (Figure 4).

Comment 3. Age of Healthy Controls: Providing information about the age at death for healthy controls is crucial, as age can impact the accumulation of aSyn. Also include if the brain samples were age-matched, or analyses were age-adjusted.

We have described the age of each patient, and the analyses were age-adjusted. Changes on page 16 (Table S1).

Comment 4. Braak Staging Discrepancy: The study reports the same Braak staging for both PD and PDD, despite the significant difference in disease duration. Maybe other reviewers with clinical experience might have a better take on this. This observation merits discussion in the paper, allowing readers to better understand the implications of this finding.

ddressed: Our PD and PDD cases are Braak stage 6, indicating that the LB pathology had progressed to the neocortex. It‘s important to note that Braak stage represents only where the LB pathogy has spread and does not indicate anything about the load of LBs. However, our immunohistochemistry results (page 20) show that PDD demonstrates a higher LB load than PD cases in the entorhinal cortex. As the reviewer has suggested, this comment has been amended in the manuscript. Changes on pages 9-11, lines 204-212.

Comment 5. Citation of Relevant Studies: The paper should consider citing and discussing a recent celebrated study on PD biomarkers that used thousands of cerebrospinal fluid (CSF) samples from different PD patient cohorts to demonstrate the effectiveness of SAA as a biochemical assay for diagnosing PD and its subtypes.

As suggested by the reviewer, we included this study in the discussion. Changes on page 12, lines 275-278.

**Reviewer 3 (Public Review):**
The experiments are missing two important controls. (1) what to fibrils generated by different in vitro fibril preparations made from recombinant synclein protein look like; and (2) the use of CSF from the same patients whose brain tissue was used to assess whether CSF and brain seeds look and behave identically. The latter is perhaps the most important question of all - namely how representative are CSF seeds of what is going on in patients’ brains?

We thank the reviewers for this valuable comment. Although in vitro preformed fibrils (PFFs) made out of recombinant aSyn are still important sources for cellular and animal studies to generate disease models and investigate mechanisms, many studies have now turned to use human brain amplified fibrils considering them to more closely present the human structure. Therefore, our study was designed to specifically address this hypothesis by comparing e human derived and SAA-amplified fibrils. It would be interesting to compare these structures also to PFFs but this was beyond the scope of our study. Comparing the CSF and brain seed from the same patients would be very interesting indeed but also difficult as this would require biosample collection during life followed by brain donation. The SAA cannot be done from the PM CSF due to contamination with blood. However, we are in a privileged position to examine such a comparison soon with our longitudinal Discovery cohort, where some participants have donated their brains. These future studies will address the critical question of whether the CSF seeds reflect those in the brain.

In their discussion the authors do not comment on the obvious differences in the conditions leading to the formation of seeds in the brain and in the artificial conditions of the seeding assay. Why should the two sets of conditions be expected to yield similar morphologies, especially since the extracted fibrils are subjected to harsh conditions for solubilization and re-suspension.

We agree with the reviewer that the formation of seeds in the brain and the SAA reaction conditions are very different, and one would not expect similar fibrillar morphologies. However, the theory is that pathological seeds are known to amplify through templated seeding, where seeds copy their intrinsic properties to the growing SAA fibrils. Thus, numerous studies use the SAA fibrils as model fibrils to investigate the different aSyn strains. Our study aimed to test whether the SAA fibrils are representative models of the brain fibrils. We included a more explicit comment on this discussion. Changes on page 3, lines 78-83.

Finally, the key experiment was not performed - would the resultant seeds from SAA preparations from the different nosological entities produce different pathologies when injected into animal brains? But perhaps this is the subject of a future manuscript.

We agree this is an essential experiment to build on our conclusion. Animal studies would be imperative to assess whether the SAA fibrils reflect the brain fibrils’ toxicity. However, these were beyond the scope of the present study but are being performed in collaboration with some expert groups.

Furthermore, the authors comment on phosphorylation patterns, stating that the resultant seeds are less heavy phosphorylated than the original material. Again, this should not be surprising, since the SAA assay conditions are not known to contain the enzymes necessary to phosphorylate synuclein. The discussion of PTMs is limited to pS-129 phosphorylation. What about other PTMs? How does the pattern of PTMs affect the seeding pattern.

We agree with the reviewer that other PTMs should be explored, but this was beyond the scope of this study. Here, we could focus on pS129, which has multiple reliable antibodies that also work with immunogold-TEM.

Lastly, the manuscript contains no data on how the diagnostic categories were assigned at autopsy. This information should be included in the supplementary material.

Clinical and neuropathological diagnostic criteria are now included in Table S1. Changes on page 16, lines 448-461.

**Reviewer 1 (Recommendations for the authors):**
(1) Remove a duplicate sentence in line 94-96.

Addressed: Thank you for pointing this out. The duplicated sentence has been corrected. Changes are on page 4, lines 105-106.

(2) Figure 1 Placement of Healthy Controls: Moving the graph representing healthy controls from the supplementary materials to the main figures could help readers better appreciate the results of diseased states.

The healthy control SAA curves were moved to the main figure. Changes are on page 5, Figure 2.

(3) Commenting on Case 2 Healthy Control: In the discussion section, you may comment on the case of the healthy control that showed amplification towards the end. While definitive conclusions may be challenging, acknowledging the possibility of incidental Lewy bodies or the prodromal phase of the disease would add depth to the analysis? But make sure to include the age information for healthy controls.

We believe this is an important point to discuss in the manuscript. We have referenced other studies with similar observations and stated that it is currently unknown what this phenomenon reflects (page 11, lines 221-226). The age information of the healthy control subjects was added to Table S1.

(4) Figure S3 Clarity: To enhance the clarity of Figure S3, consider adding a reference marker or arrow in the low-magnification image that points to the region being magnified in the insets. This visual cue will make it easier for readers to connect the detailed insets with the corresponding area in the broader image.

In Figure S3, we included a reference arrow in the low-magnification images to clarify where the higher-magnification images are taken. Changes are on page 19, Figure S3.

**Reviewer 2 (Recommendations for the authors):**
(1) A major issue confronting the field is the conflation of the PMCA and RT-QuIC assays (the latter of which was used here). The decision to rename and combine the two under the umbrella of SAAs does a major disservice to the field for many reasons. Recognizing that the push for this did not come from the authors, clarifying the differences in their Introduction would be very useful. I suggest this, in large part, because in the prion field, PMCA is known to amplify prion strains with high fidelity whereas the product from RT-QuIC does not. In fact, the RT-QuIC product for PrP is not even infectious, while the synuclein field uses it as a means to generate material for subsequent studies. Highlighting these differences would certainly strengthen the arguments the authors are making about the inadequacy of the synuclein RT-QuIC approach in research.

We thank the reviewers for these very valuable comments. We have included a further introduction on PMCA and RT-QuIC, explaining the differences and clearly stating our selection of the RT-QuIC method in this paper (page 3, lines 55-68). In addition, we have highlighted that, unlike PMCA, the RT-QuIC end-products are non-infectious and biologically dissimilar to the seed protein. Combined with our results, the findings demonstrate the methodological limitation of RT-QuIC in reproducing the seed fibrils and replicating their intrinsic biophysical information.

(2) On page 4, sentences starting on lines 94 and 95 are a duplication.

The duplicated sentence has been corrected. Changes are on page 4, lines 105-106.

(3) In the Results, noting that the pSyn staining on the RT-QuIC fibrils is coming from the human patient sample used to seed the reaction would be useful. This is mentioned in the Discussion, but the lack of mention in the Results made me pause reading to double check the methods. I think this could also be addressed a bit more clearly in the Abstract.

We have clarified this in the Results and Abstract. Changes on page 1 (lines 21-22) and page 9 (lines 192-194)

(4) On page 8 line 188, change was to were in the sentence, ”First, faster seeding kinetics was...”

This grammar error has been corrected. Changes are on page 9, line 200.

(5) The authors may want to comment on the unexpected finding that despite the RT-QuIC fibrils having a difference in twisted vs straight filaments, all 4 seeded reactions gave identical results in the conformational stability assay.

Addressed: We want to thank the reviewer for this comment and have highlighted the unexpected finding with a comment on what could be causing the identical results in the conformational stability assay. Changes are on page 12, lines 297-303.